# Cadaverine as a Potential Spoilage Indicator in Skin-Packed Beef and Modified-Atmosphere-Packed Beef

**DOI:** 10.3390/foods12244489

**Published:** 2023-12-15

**Authors:** Kristian Key Milan Thamsborg, Birgitte Winther Lund, Derek Victor Byrne, Jørgen Johannes Leisner, Niki Alexi

**Affiliations:** 1Department of Veterinary and Animal Sciences, University of Copenhagen, Grønnegårdsvej 15, 1870 Copenhagen, Denmark; 2Danish Technological Institute (DMRI), Gregersensvej 9, 2630 Taastrup, Denmark; 3Food Quality Perception and Society Science Team, iSENSE Lab, Department of Food Science, Faculty of Technical Sciences, Aarhus University, Agro Food Park 48, 8200 Aarhus N, Denmark; derekv.byrne@food.au.dk (D.V.B.); niki.alexi@food.au.dk (N.A.)

**Keywords:** shelf life, meat quality, lactic acid bacteria, LC-MS/MS, biogenic amine, freshness, food packaging

## Abstract

This study investigated cadaverine as a spoilage indicator in commercial beef products stored under conditions favourable for the growth of lactic acid bacteria. Samples included vacuum-skin-packed entrecotes (EB) aged up to 42 days and modified-atmosphere-packed (70% O_2_ + 30% CO_2_) minced beef (MB) stored at 5 °C. Two MB product lines were analysed: one stored aerobically two days post-slaughter before mincing and another stored for 14 days in vacuum packaging prior to mincing. Sensory assessment/evaluation and microbial analysis were performed throughout the shelf life of the products and compared to cadaverine levels measured using LC-MS/MS. Cadaverine concentrations in EB reached approximately 40,000 µg/kg on the “best before” date, while remaining below 50 µg/kg in both MB products on the corresponding date. While cadaverine concentrations in EB displayed a consistent increase, suggesting its potential as a spoilage indicator post-ageing, the low concentrations in MB, did not correlate with sensory assessments, revealing its limitations as a universal spoilage marker. In conclusion, it is necessary to conduct product-specific studies to evaluate the applicability of cadaverine as a spoilage indicator for beef products.

## 1. Introduction

Around one-third of global food production is lost due to spoilage. This has an environmental impact due to the depletion of food resources [1]. Up to 10% of Europe’s annual food waste is associated with misconceptions regarding date marking, regardless of actual quality [2,3]. As beef has a significant waste footprint, more precise and effective shelf-life predictions will generate a positive environmental outcome [4].

Cadaverine is a biogenic amine generated by decarboxylation of lysine. Concentrations in meat are determined by microbial enzymatic activities as well as characteristics of the meat tissue [5,6]. An increase in cadaverine levels in beef during storage has been observed under various conditions, making it a potential alternative to traditional spoilage indicators [7,8,9,10,11]. As a result, rapid methods like novel biosensors to measure cadaverine have been developed [12,13,14]. 

The applicability of cadaverine as a quality and shelf-life metric depends on microbial composition and growth. Enterobacterales contamination and growth in beef is often regarded as a likely source of cadaverine, but this depends on packaging conditions [7,8]. Thus, vacuum packaging (VP) and modified atmosphere packaging (MAP) select for lactic acid bacteria (LAB), with high CO_2_ levels in MAP inhibiting Gram-negative bacteria, including Enterobacterales [15,16]. Most studies on cadaverine in beef relate to products stored under vacuum or aerobic conditions [8,10,17,18,19,20]. There is, however, limited research comparing cadaverine levels to microbial and sensory data during storage of ageing beef in VP. This also applies to beef stored in MAP before mincing. Both product types represent common storage procedures in Danish meat production. This study aims to evaluate the utility of cadaverine as a spoilage indicator in these types of products. 

## 2. Materials and Methods

### 2.1. Samples and Packaging

Entrecotes (EB) and minced beef (MB) were obtained from Danish Crown A/S (Randers, Denmark) and originated from animals reared on conventional Danish farms. The EB was stored as striploins for up to 42 days at 5 °C, before being cut and vacuum skin packaged (Figure 1). This is a standard ripening process in the meat industry and was selected as an alternative to the minced products. The shelf-life assessment period for the EB was 25 days after packaging (D1 to D25) at 5 °C, with a “best before” date assigned by the producer on D21. The MB portions, which weighed 500 g and had a fat content of 10–12%, were packed in MAP with 70% O_2_ + 30% CO_2_. The shelf-life assessment period for the MB was eight days at 5 °C after packaging, with a “best before” date at D7 according to the standard determined by the producer (Figure 1). Two MB products of varying freshness were examined: one stored for two days post-slaughter before mincing (MB0) and one that had been stored for fourteen days in VP at 5 °C prior to mincing (MB14). The duration was based on realistic scenarios in the meat industry, since products that are stored for up to fourteen days in VP can acquire the same expiry date as products obtained from fresh beef minced only two days post-slaughter. Thus, it is of interest to explore cadaverine in such scenarios, to examine whether it is possible to discriminate these small variations in quality. At each sampling day, three replicate product packages of EB, MB0, and MB14, respectively, were used for sensory evaluation. For cadaverine and microbial analyses, each was conducted with two separate product packages in duplicates, leading to a total of four measurements on designated days (Figure 1). This resulted in three, five, or seven replicate product packages being used to evaluate each of EB, MB0, and MB14 depending on the number of analyses performed on that sampling day. 

### 2.2. Sensory Evaluation

An internal, trained panel of 5–8 judges at the iSENSE Lab (Department of Food Science, Faculty of Technical Sciences, Aarhus University) assessed sensory quality in triplicate on designated days (Figure 1), based on odour and appearance (colour shifts) using printed schemes. The odour scale categories used for the EB were “raw meat odour (bloody, metallic, fresh)”, “raw meat odour with some rancid buttery, fishy notes”, “sulphuric (eggy), fishy, rancid with raw meat notes”, or “eggy, fishy, rotten”, whereas a five-point scale ranging from “fresh raw meat” to “putrid/decaying odour” was used for the MB, following industry standards. The products’ colours were categorised as “Bright red”, “Dark red”, or “Dark brown”. Additionally, the panel was asked if the product sample was acceptable for consumption (Yes/No). The end of shelf life was marked by the first day on which a panellist considered one replicate to be “unacceptable”.

### 2.3. Microbial Evaluation

Microbial counts for both products were obtained in proximity to the “best before” dates. Two replicate product packages were analysed in duplicate. EB counts were obtained on D18 and D21–25, and for MB on D5–8 (Figure 1). Samples (10 g) were homogenised in 0.1% peptone saline (CM0982; Oxoid Ltd., Basingstoke, UK) using a stomacher (Stomacher 400 Lab Blender, Seward Medical, London, UK) (2 min, highest speed) and relevant serial dilutions were performed. Total viable counts (TVCs) were obtained using plate count agar (PCA; Oxoid) (25 °C, 5 d), psychrotrophic counts on Long and Hammer (LH) medium (15 °C, 7 d), and LAB counts on nitrite polymyxin (NP) medium stored under microaerophilic conditions (25 °C, 3 d). NP medium consisted of APT agar (Merck, Darmstadt, Germany), with 1 mL of 12% sodium nitrite (Ampliqon A/S, Odense, Denmark) and polymyxin supplement (Oxoid, SR 099) added per 200 mL agar. *Brochothrix thermosphacta* counts were obtained on streptomycin thallous acetate (STA, Oxoid) medium (25 °C, 3 d). Enterobacterales, including coliforms and *E. coli*, were enumerated using Rapid’Enterobacteriaceae (Rapid’Entero; Biorad, Hercules, CA, USA) (25 °C, 2 d) and Rapid *E. coli* 2 (Biorad) (37 °C, 1 d) media. Forty (MB) and thirty (EB) randomly selected isolates from PCA plates evenly distributed between sampling days were further analysed using MALDI-TOF MS (Biomérieux, Craponne, France) after cultivation on tryptone soya agar (Oxoid) with 5% pig blood, as well as subjected to 16s rDNA identification by Macrogen (Europe) if required. 

### 2.4. Cadaverine

Cadaverine concentrations were measured using LC-MS/MS with electrospray ionisation. EB samples were analysed on D1, D8, D11, D18, D21, and D23, and the MB on D1 and D4–8 (Figure 1). Two replicate product packages were analysed in duplicate and stored at −80 °C prior to simultaneous analysis. A 2 g blended subsample was homogenised with 30 mL of 0.1 HClO_4_ in a PP centrifuge tube, by mixing for 30 s at 2000 rpm, and shaking for 15 min. The sample was then centrifuged (4500 rpm, 60 min, 4 °C). Cadaverine extraction was performed using perchloric acid, as previously outlined [12,21,22]. Subsequent purification used Oasis WCX (60 mg, 3 cc) (Waters Corp., Milford, MA, USA) and Strata XL-CW (100 mg, 6 cc) (Phenomenex, Torrence, CA, USA) ion-exchange columns for MB and EB, respectively [23], followed by chromatographic separation using a reverse-phase column (Kinetex 2.6 µm C18 100A, 100 × 3 mm) (Phenomenex) with heptafluorobutyric acid as an ion-pairing agent [24,25]. Detailed descriptions of each step can be found in [12]. The calibration process followed [12], with cadaverine dihydrochloride (CAS no. 1476-39-7) (Merck) measurements before and after spiking. The limit of quantification (LOQ) for both matrices was 20 µg/kg. 

### 2.5. Statistical Analyses

Statistical analyses evaluated differences in sensory quality scores, microbial counts, and cadaverine concentrations across storage days. Two-way ANOVA was used to assess the relationship between product and storage for MB0 and MB14, while one-way ANOVA was used for EB. Analyses employed R with agricolae for the post hoc Duncan test [26,27]. Linear regression, Pearson correlation (*p* value: two-tailed, CI: 95%), and visualisation were performed using GraphPad Prism version 10.0.1 (GraphPad Software, www.graphpad.com accessed on 3 August 2023). Results are presented as means and standard error (SE), with relative comparisons when appropriate.

## 3. Results and Discussion

### 3.1. Sensory Evaluation

Sensory evaluation of MB0, MB14, and EB revealed notable changes in colour and odour during storage (Figure 2, Figure 3 and Figure 4). Since both MB0 and MB14 were considered “acceptable for consumption” until D8, an additional sensory evaluation was performed on D11. Interestingly, only MB14 was found unacceptable in sensory evaluation on D11 (Table 1), consequently marking the end of shelf life as between D9 and D11 for MB14. In conclusion, the shelf life could be extended by two to four days for MB14 and by at least four days for MB0, when compared to the “best before” date (D7) determined by the producer. The shelf life was longer than anticipated for both MB products which underscores the importance of the correct estimation of shelf life. The EB was deemed unacceptable for consumption on D23, based on sensory evaluation, close to the “best before” date (D21).

Both MB products started with a “bright red” appearance, turning “dark red” by D5 (Figure 2). This colour remained consistent from D5 to D8, signifying no changes in either product. However, on D11, MB14 transitioned to a “brown colour”, confirming a significant shift in quality (*p <* 0.01). In contrast, MB0 retained a “dark red” colour on D11, showing no difference from D8 (*p* = 0.17). The colour of EB was initially classified as “red” and gradually transitioned to “dark red”, which was consistent from D23 (Figure 4).

No discrimination in odour quality between the two MB products was evident on D1 and D4 (Figure 3). Both products exhibited a noticeable odour transition from “fresh raw meat” to “raw meat with some lactic acid/ buttery notes” during this period. From D5, MB0 was perceived as significantly fresher than MB14 (*p* < 0.01). MB0 maintained the D4 odour profile until D8, while MB14 transitioned to a less fresh odour, described as “lactic acid buttery, with some raw meat notes”, which could be associated with MB14 having LAB counts that were approximately two log units higher. On D11, both products had significantly less fresh odours than on D8 (*p <* 0.01). A linear increase in odour during storage was more obvious for MB14 than for MB0 (Figure 3). The same linearity would probably exist for MB0 if more samples later in the shelf life were obtained. EB sensory quality showed a good correlation with storage (Figure 4). Initially, the odour in EB was generally perceived as “fresh raw meat”, but it transitioned to a state between “raw meat with rancid notes” and “rancid with raw meat notes” by D23. Despite not reaching a brown colour or putrid smell, the transition in colour and odour around the end of the shelf life indicates a definitive change in the EB. The colour stability in VP aligns with the literature, while odour proves a more sensitive spoilage indicator during storage [28]. It is noteworthy that while changes in odour quality were more variable and discriminative for quality, a transition to “dark brown” led to immediate product rejection, highlighting the importance of colour for perceived product expiry and consumer rejection [29].

### 3.2. Microbial Evaluation

Differences in bacterial counts were observed between the two MB products (Table 1). Both displayed a TVC increase of around 1.5 log units from D5 to D8, reaching around log 5.8 CFU/g (SE = 0.1) for MB0 and log 8.4 CFU/g (SE = 0.0) for MB14. Similar trends were observed for LAB and psychrotrophic counts. Rapid’Entero counts increased during storage and reached log 4.4 CFU/g (SE = 0.1) for MB14 and log 2.9 CFU/g (SE = 0.0) for MB0. Some Rapid’Entero and coliform counts were estimates, due to the low numbers of colonies when using the lowest dilution, and *E. coli* was only detected in a very few cases. Counts of *Brochothrix thermosphacta* in both MB products showed an increase of two log units during storage. TVC, LAB, and psychrotrophic counts for EB reached a plateau around log 8 CFU/g (SE: 0.1–0.3) from D18. Rapid’Entero reached a peak of log 6.2 CFU/g (SE = 0.2) on D24 and was significantly higher at the “best before” date when compared to the MB products (*p* < 0.01, Table 1). Coliforms peaked at ~log 2.5 CFU/g (SE = 0.3), while *Brochothrix thermosphacta* were below log 2 CFU/g. Notably, microbial spoilage thresholds of log 7 CFU/g [30] were reached well before the “best before” date for both MB14 and EB. The predominance of LAB and psychrotrophic bacteria was consistent with previous findings [28,30,31], suggesting that these products represented typical bacterial spoilage of beef stored in VP and MAP with high CO_2_ at low temperatures. *Brochothrix thermosphacta* were present in higher numbers in MB products, despite the high content of CO_2_, which did not agree with previous findings [31]. 

Variation in bacterial genera among the isolates from PCA plates was greater in MB0 compared to MB14 and EB (Table 2). Due to the limitations of MALDI-TOF when applied to identify the specific taxa of LAB, especially *Leuconostoc* spp., selected isolates were identified using 16s rDNA sequencing. Gram-positive/catalase-negative ovococcoid that formed chains resembling leuconostocs were annotated as *Leuconostoc* spp. LAB predominated in all products, with relatively similar presences of *Carnobacterium* spp., *Latilactobacillus* spp., and *Leuconostoc* spp. in MB0, while the latter genus dominated MB14 and EB. The dominance of LAB aligns with the assertion that LAB tends to dominate during storage in VP and MAP [16], while growth of Enterobacterales and pseudomonads is known to be inhibited or limited under such conditions [15]. Enterobacterales isolates were obtained from MB0 and EB samples, with 10% of EB isolates identified as *Serratia proteamaculans*, which indicates relatively high numbers. *S. proteamaculans* is known for its potential to produce cadaverine in beef and therefore could potentially be responsible for the production of cadaverine in this product [32]. Rapid’Entero isolates from the MB products included *Serratia liquefaciens*, *Hafnia alvei*, and *Pseudomonas fragi*. *Hafnia* spp. and *Pseudomonas* spp. are also known as cadaverine producers [33,34]. However, high CO_2_ concentrations may limit the growth of Enterobacterales and the production of cadaverine, which would minimise the potential of cadaverine as a spoilage indicator in MB products [16,35].

### 3.3. Cadaverine

Cadaverine levels in the MB products were around LOQ (Figure 5), but higher in EB (Figure 6), in agreement with studies on vacuum-stored beef [10,17,34]. Cadaverine concentrations reached levels tenfold higher than the levels measured in pork cutlets stored under MAP at 5 °C [36], but well below the lowest proposed maximum (430,000 µg/kg) in food products [37]. 

Cadaverine levels in EB showed an exponential relationship with storage time (Figure 6). Concentrations of cadaverine in MB0 and MB14 samples did not fit exponential or linear models very well, although obtaining additional measurements further into the shelf life and closer to spoilage may have indicated a pattern as seen in other meat and fish products [12,36]. Cadaverine concentrations seemed to decrease at D8 (Figure 5). However, a similar decrease was observed for odour in both MB products, followed by an increase thereafter (Figure 3). A positive correlation between cadaverine and odour was observed for MB14 (r = 0.94, *p* < 0.01), while the correlation was not significant for MB0 (r = 0.70, *p* = 0.12). The analysis indicates that cadaverine is a potential spoilage indicator in the MB products, with some limitations. The dominant genera identified in this product are not known as strong producers of cadaverine [6,38]. Known cadaverine-producing bacteria like *Serratia* spp. and *Hafnia* spp. most likely were not present in sufficiently abundant concentrations to generate significant levels of cadaverine. Despite the overall low cadaverine concentrations observed in the MB products, ANOVA results still indicated a significantly (*p* = 0.03) higher concentration in MB0 when compared to MB14. This could be due to the greater diversity found in MB0, potentially due to pre-mincing aerobic storage. However, if cadaverine should be used as an indicator of quality, MB14 would be expected to have higher concentrations than MB0, based on higher bacterial counts, presence of off odours, as well as a shorter shelf life indicated by the experiment. Furthermore, observed TVCs above legislative standards for a product at the end of shelf life (>log 7.5 CFU/g, Danish regulation nr. 9774, 30/6/22) at D5/6 (MB14) and sensory rejection at D11 showed a discrepancy regarding the duration of shelf life. This emphasises the potential effects of pre-product processes on the usability of cadaverine as a spoilage indicator. 

The increase in cadaverine in EB underscores its potential as a spoilage indicator in this type of product. Rapid cadaverine measurements could be a cost-effective method to ensure product freshness at the factory level prior to distribution, as well as at the retail level upon receipt of the products. However, further research is required to utilise cadaverine concentrations to obtain an accurate prediction of expected shelf life. Products with slow growth of Enterobacterales would potentially make ideal candidates for the application of cadaverine as a quality parameter, as this scenario may lead to a gradual increase in cadaverine over the entire shelf life. This scenario is more promising than the situation where cadaverine only significantly increases after extensive spoilage has occurred [39].

## 4. Conclusions

This study revealed that levels of cadaverine could not be applied to differentiate the quality of the different minced beef products, as opposed to sensory odour and microbial analyses. Moreover, the generally low cadaverine levels in the minced beef posed a challenge to obtaining accurate measurements. In contrast, a steady increase in cadaverine in the entrecotes suggests its potential as a spoilage indicator in skin-packed beef post-aging. The differences in cadaverine concentration throughout the storage period between these products revealed its limitations as a general spoilage marker. These results emphasise the importance of product-specific investigation if cadaverine is to be considered as a spoilage indicator for beef products.

## Figures and Tables

**Figure 1 foods-12-04489-f001:**
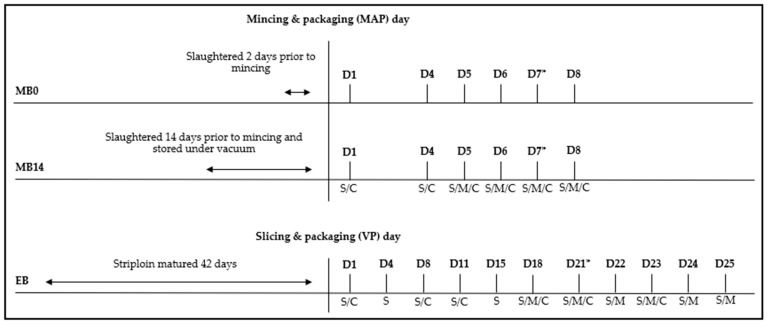
Setup of the study, showing sampling days (D) along with the analysis performed on that specific sampling day (S: sensory evaluation/M: microbial analysis/C: cadaverine analysis). Abbreviations: minced beef (MB); entrecotes (EB); modified atmosphere packaging (MAP); vacuum packaging (VP); * best before date assigned by the producer.

**Figure 2 foods-12-04489-f002:**
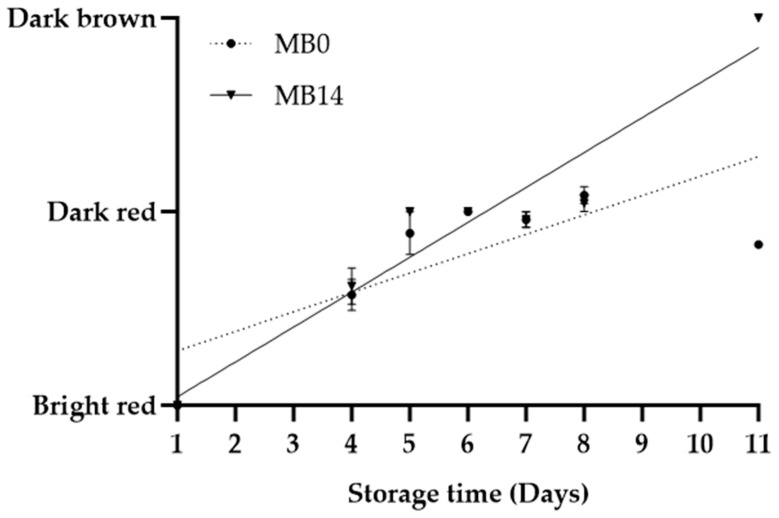
Colour changes in minced beef stored for 2 days prior to mincing (MB0), or for 14 days prior to mincing in VP (MB14), when stored at 5 °C. Values are presented as mean ± SE; dotted and solid lines represent the linear relationships for MB0 (R^2^ = 0.58) and MB14 (R^2^ = 0.90), respectively.

**Figure 3 foods-12-04489-f003:**
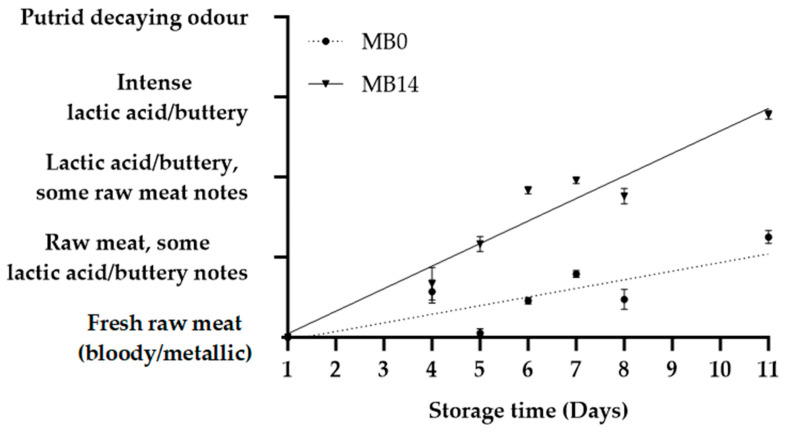
Odour changes in minced beef stored for 2 days prior to mincing (MB0), or for 14 days prior to mincing in VP (MB14), when stored at 5 °C. Values are presented as mean ± SE; dotted and solid lines represent the linear relationships for MB0 (R^2^ = 0.59) and MB14 (R^2^ = 0.91), respectively.

**Figure 4 foods-12-04489-f004:**
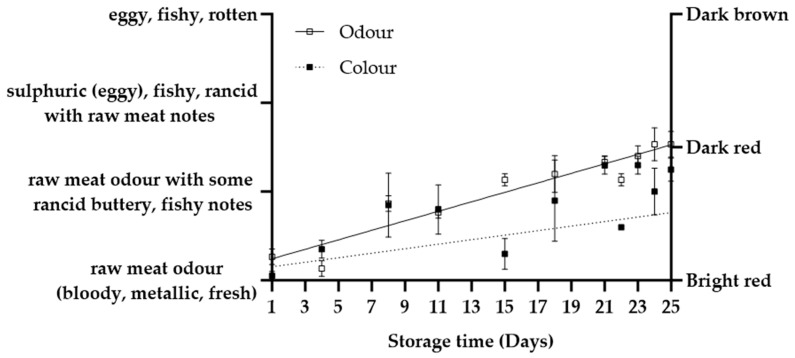
Odour and colour changes in entrecotes beef (EB) stored at 5 °C. Values are presented as mean ± SE; dotted and solid lines represent the linear relationships for odour (R^2^ = 0.80) and colour (R^2^ = 0.35), respectively.

**Figure 5 foods-12-04489-f005:**
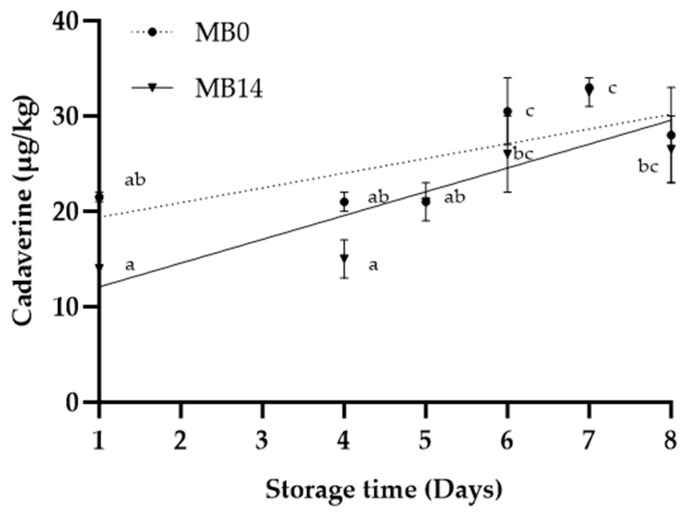
Cadaverine concentrations measured via LC-MS/MS in minced beef stored for 2 days prior to mincing (MB0), or for 14 days prior to mincing in VP (MB14), when stored at 5 °C. Values are presented as mean ± SE; lowercase letters indicate post hoc groupings according to the Duncan test performed; dotted and solid lines represent the linear relationships for MB0 (R^2^ = 0.41) and MB14 (R^2^ = 0.65), respectively.

**Figure 6 foods-12-04489-f006:**
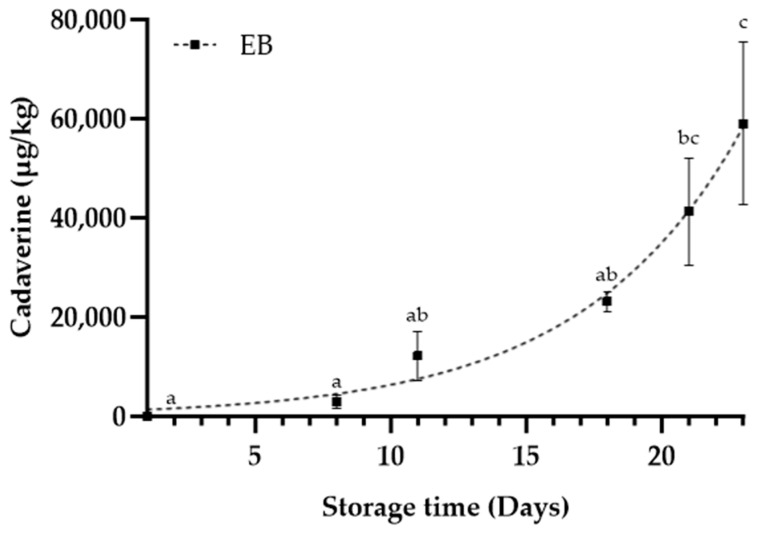
Cadaverine concentrations measured via LC-MS/MS in entrecotes beef (EB) stored at 5 °C. Values are presented as mean ± SE; lowercase letters indicate post hoc groupings according to the Duncan test performed; the dashed line represents an exponential relationship for EB (R^2^ = 0.86).

**Table 1 foods-12-04489-t001:** Sample characteristics and results at the “best-before” date.

	MB0	MB14	EB
**Sample characteristics**			
*Ageing*	2 days at 5 °C	VP: 14 days at 5 °C	42 days at 5 °C
*Processing (D1)*	Minced	Minced	Entrecote
*Packaging*	MAP: O_2_ 70% + CO_2_ 30%	MAP: O_2_ 70% + CO_2_ 30%	VP
*“Best before” date ^a^*	D7	D7	D21
*Unacceptable ^b^*	>D11	D11	D23
**“Best-before” date ^a^ results**			
*Colour*	Dark red	Dark red	Dark red
*Odour*	Raw meat, some lactic acid/buttery notes	Lactic acid/buttery, some raw meat notes	Raw meat, some rancid buttery, fishy notes
*Total viable counts*	5.0 ± 0.1	7.9 ± 0.1	8.1 ± 0.0
*Psychrotrophic counts*	5.4 ± 0.1	7.7 ± 0.2	8.2 ± 0.1
*Lactic acid bacteria*	5.6 ± 0.2	8.3 ± 0.1	8.1 ± 0.2
*Brochothrix thermosphacta*	3.7 ± 0.1	6.2 ± 0.1	<2
*Rapid’Enterobacteriaceae ^c^*	2.9 ± 0.2	4.1 ± 0.0	4.9 ± 0.1
*Coliforms*	1.6 ± 0.1	2.1 ± 0.2	<1
*Cadaverine (µg/kg)*	~33 ± 0	~33 ± 2	~40,000 ± 11,000

Data from two replicate packages evaluated in duplicate; values are given as means ± SE; bacterial counts in log CFU/g; minced beef (MB); entrecotes (EB); vacuum packaging (VP); modified atmosphere packaging (MAP); day (D); ^a^ determined by the producer; ^b^ the first day a panellist considered one of the samples unacceptable according to sensory evaluation; ^c^ Pseudomonads were isolated from these plates along with Enterobacterales.

**Table 2 foods-12-04489-t002:** Identification of plate count agar isolates using MALDI-TOF or 16S rDNA analysis.

Taxon	MB0	MB14	EB
*Brochothrix thermosphacta*	1	1	
*Burkholderia lata*	1		
*Carnabacterium divergens*	7	3	4
*Carnobacterium maltaromaticum*	1		
*Hafnia alvei*	1		
*Kocuria rhizophila*	1		
*Lactococcus piscium* *	2		
*Lalilactobacillus fuchuensis*	1		3
*Latilactobacillus sakei*	6	1 *	
*Leuconostoc carnosum*		2	
*Leuconostoc gasicomitatum* *	7	6	
*Leuconostoc gelidum* *	1	6	
*Leuconostoc mesenteroides*	1		
*Leuconostoc pseudomesenteroides*		1	
*Leuconostoc* spp. **	4	13	14
*Pseudomonas fragi*	2		
*Pseudomonas* sp.	1		
*Serratia liquefaciens*	1		
*Serratia proteamaculans*			3
*Fusarium chlamydosporum*		1	
*Pseudallescheria boydii*	1	5	4
Unidentified	1	1	2
Number of isolates	40	40	30

Minced beef stored for 2 days prior to mincing (MB0), or for 14 days prior to mincing in VP (MB14); entrecotes beef (EB). * Identified by 16S rDNA. ** Isolates presumptive of *Leuconostoc* spp., based on phenotype and microscopy.

## Data Availability

Data is contained within the article.

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
