# Peer review of "Cadaverine as a Potential Spoilage Indicator in Skin-Packed Beef and Modified-Atmosphere-Packed Beef"

_foods, 2023, doi:10.3390/foods12244489_

Round 1

Reviewer 1 Report

Comments and Suggestions for Authors

The work fits the journal scope.

line 51. 2.1 Samples and Packaging. Entrecotes and minced beef samples were used. The meat samples were stored-packed-stored the different ways and tested at different days of storage. I suggest add a graphical "sampling design" so that a Reader could address the picture and understand better which sample is which.

You wrote (lines 63-65): Depending on the analyses, three, five, or seven sample replicas were evaluated for each of EB, MB0, and MB14 at each sampling point. Please specify the number of replicas evaluated for other EB and MB samples.

Please explain how samples D21-25 (for EB) and  D4-8 (for MB) were prepared for testing.

You wrote that the "best before" day for EB is D21 and D7 for MB. But you did not analyze samples on these days. Please explain.

line 67 2.2 Sensory Evaluation. There are no data in Table 1 that confirm the results obtained and conclusions made. The statement that minimum shelf-life extension of two days for MB14 and four for MB0 seems not be fully supported with the results from the Table 1.

To follow the Authors' explanation and conclusions in the Results and Discussion part more data needed. In a Supplementary, perhaps.

The paper discusses the potential of just one biogenic amine - cadaverine, as a marker of meat spoilage.

Manuscript might be of interest to a limited number of specialists.

Author Response

Ms. Ref. No.: foods-2750771

Title: Cadaverine as a Potential Spoilage Indicator in Skin-Packed Beef and Modified Atmosphere-Packed Beef

Response to reviewer #1:

General comment

The work fits the journal scope.

We want to thank the reviewer for taking the time to review and deliver useful feedback for our manuscript. We have addressed all the reviewer’s comments in this letter and the revised manuscript, and we believe it has improved the manuscript greatly. Furthermore, we have done a full review of the manuscript and added some minor corrections to further improve the content and quality. These changes can be found under author corrections last in this letter, the changes are mainly editorial. The changes within the manuscript are indicated in red font, and the response to the reviewer comments are included below in italics.

Material and methods

line 51. 2.1 Samples and Packaging. Entrecotes and minced beef samples were used. The meat samples were stored-packed-stored the different ways and tested at different days of storage. I suggest add a graphical "sampling design" so that a Reader could address the picture and understand better which sample is which.

We thank the reviewer for suggesting this improvement, we agree that it would help the reader and improve the readability. We have produced a figure showing the sampling design for the reader. The figure was added after section 2.1 Samples and Packaging along with its figure caption. Finally, we changed figure references accordingly in the whole manuscript.

Line 56: “(Figure 1)” added after “packaged”

Line 82: “(EB: D1, D4, D8, D11, D15, D18, & D21-25; MB: D1 & D4-8)” replaced by “(Figure 1)”

Line 95: “(Figure 1”) added after D5-8

Line 115: “(Figure 1)” added after D4-8.

You wrote (lines 63-65): Depending on the analyses, three, five, or seven sample replicas were evaluated for each of EB, MB0, and MB14 at each sampling point. Please specify the number of replicas evaluated for other EB and MB samples.

We believe a miscommunication is the reason for this comment, and we have clarified it by referring to Figure 1 (see prior comment) and changed  lines 63-65 to the following:

Lines 68-71: “At each sampling day, three replicates of each EB, MB0, and MB14 were used for sensory evaluation. For cadaverine and microbial analyses, two separate tests were conducted on each of two replicates, leading to a total of four measurements on designated days (Figure 1).”

Please explain how samples D21-25 (for EB) and D4-8 (for MB) were prepared for testing.

We thank the reviewer to point out the confusion related to this part of the text. We believe that the addition of figure 1 have improved the methods section in way it is easier to understand. The explanation was that EB and both MB products were analysed by sensory evaluation on all these days in in triplicates. Instead, we are now referring to the sample design figure, so the reader will look at it and not get confused.

Line 82: “(EB: D1, D4, D8, D11, D15, D18, & D21-25; MB: D1 & D4-8)” replaced by “(Figure 1)”

You wrote that the "best before" day for EB is D21 and D7 for MB. But you did not analyze samples on these days. Please explain.

This is a valid observation; we agree that additional information is needed to know this. The “best before” date was determined by the producer of the product. We did analyse the samples on the respective days (sensory evaluation, microbial counts, and cadaverine content), we would like to refer to the sample design figure that the reviewer was kind to suggest us to make. The following comment has been added to the manuscript.

Lines 58-59: “assigned by the producer” was added before “at D21”

Lines 61-62: “according to the standard determined by the producer (Figure 1)” was added after “at D7”

Results and Discussion

line 67 2.2 Sensory Evaluation. There are no data in Table 1 that confirm the results obtained and conclusions made. The statement that minimum shelf-life extension of two days for MB14 and four for MB0 seems not be fully supported with the results from the Table 1.

We want to thank the reviewer for pointing out this lack of information in the manuscript, the following changes have been made to clarify what our conclusion was based on:

Lines 142-143: “by sensory evaluation on D11, consequently marking end of shelf life between D8 and D11 for MB14” added after “found unacceptable”

Lines 143-145: ”In conclusion the shelf life could be extended two to four days for MB14 and at least four days for MB0, when compared to the “best before” date (D7) determined by the producer (Table 1).” added after the previous sentence.

Lines 145-146:”The shelf-life” replaced “This” & “for both MB products” added after “anticipated”.

Lines 147-148: “The EB was deemed unacceptable for consumption on D23 based on sensory evaluation, close to the “best before” date (D21).” added after “shelf-life.”

To follow the Authors' explanation and conclusions in the Results and Discussion part more data needed. In a Supplementary, perhaps.

We agree with the reviewer that the manuscript would benefit of more data being visualised. To ensure this we have added 3 additional figures on the subject of sensory evaluation. Furthermore, we have seperated the old figure 2 (now figure 5 & 6), for easier comparison to the three new figures. Lastly we have added a Table 2 about isolate identification. We have changed the figure references accordingly.

 Line 182: Figure 2 was added along with figure caption

Line 187: Figure 3 was added along with figure caption

Line 192: Figure 4 was added along with figure caption

Line 234: Table 2 was added along with table caption

Line 246: Figure 5 was added along with figure caption

Line 252: Figure 6 was added along with figure caption

The paper discusses the potential of just one biogenic amine - cadaverine, as a marker of meat spoilage.

Manuscript might be of interest to a limited number of specialists.

We agree that it is currently so for a limited number of specialists but believe it can be important for the whole meat industry in the near future. The recent advantages in cadaverine measurements by biosensors and colorimetric approaches within the last half a decade emphasize the importance of studies specifically focused on cadaverine as a spoilage indicator.

Reviewer 2 Report

Comments and Suggestions for Authors

Abstract

Lines 12-13: Please indicate here if skin packed entrecotes (EB) means that they were vacuum-packaged

Line 15: Explain VP abbreviation

Material and methods:

2.1. Subheading. The rationale for the experimental design should be explained. What was pursued with the different ripening, packaging and storage regimes for the samples? It is understood that the end of shelf life was determined by panelists based on sensory analysis, but what was the determining factor for the best-before date: sensory, microbial count or chemical decomposition (e.g TBARS values)?

Line 69: What is AU short for? Please explain all abbreviations at first use in Abstract, Manuscript, Figures and Tables.

Milliliter is written as ml (lines 90 and 91) or mL (line 104). Please standardize the use of abbreviations.

Line 113-114: Do you mean the LOQ for both matrices (not for both methods)

Even though the LC-method for cadaverine is described elsewhere by the authors, please indicate here the retention time.

Results and discussion

Line 171: The LOQ for Rapid’entero and coliform has not been previously defined. The terms LOQ and LOD are not widely used in microbiology to describe the minimum number of microbes that can be detected in a given analytical approach.

Author Response

Response to reviewer #2:

General comment

We want to thank the reviewer for taking the time to review and deliver useful feedback for our manuscript. We have addressed all the reviewer’s comments in this letter and the revised manuscript, and we believe it has improved the manuscript greatly. Furthermore, we have done a full review of the manuscript and added some minor corrections to further improve the content and quality. These changes can be found under author corrections last in this letter, the changes are mainly editorial. The changes within the manuscript are indicated in red font, and the response to the reviewer comments are included below in italics.

Abstract

Lines 12-13: Please indicate here if skin packed entrecotes (EB) means that they were vacuum-packaged

We thank the reviewer for pointing out this mistake, it is changed accordingly.

Line 12: “vacuum” added after “included”

Line 15: Explain VP abbreviation

We thank the reviewer for commenting on this missing description, it has been changed in the manuscript.

Line 15: “VP” changed with “vacuum packaging.”

Material and methods

2.1. Subheading. The rationale for the experimental design should be explained. What was pursued with the different ripening, packaging and storage regimes for the samples? It is understood that the end of shelf life was determined by panelists based on sensory analysis, but what was the determining factor for the best-before date: sensory, microbial count or chemical decomposition (e.g TBARS values)?

We agree with the reviewer that further explanation behind the products chosen is needed. The following has been added:

Line 56-57: “This is a standard ripening process in the meat industry and was selected as an alternative to the minced products.” added after “(Figure 1).”

Line 64-68: “The duration was based on realistic scenarios in the meat industry, where tissue stored up to 14 days in VP can acquire the same expiry date as mince obtained from fresh beef minced only two days post-slaughter. Thus, it was very interesting to explore cadaverine in such scenarios to see if it could discriminate these small variations in quality.”  added after “(MB14).”

We thank the reviewer for the observation on the “best before” date, we agree that additional information is needed. The following comments has been added to the manuscript:

Line 59: “assigned by the producer” was added before “at D21”

Line 62: “according to the standard determined by the producer” was added after “at D7”

Line 152: “aDetermined by the producer” added after “Day(D);”

Line 153: “according to sensory evaluation” added after “unacceptable”

The following has been changed to clarify the overall reasoning to choose these products in the introduction (line 39-48):

Line 42: added “and modified atmosphere packaging (MAP)” after “(VP)”

Line 43: “while” replaced by “and” & “modified atmosphere packaging” was deleted.

Line 45: “, however,” was added after “is”

Line 47: “and” replaced by “. This also apply to”

Line 48: “, both” replaced by “. Both product types represent”

Line 69: What is AU short for? Please explain all abbreviations at first use in Abstract, Manuscript, Figures and Tables.

We appreciate the reviewer for noticing this omission, it has been changed accordingly. We have looked through the manuscript and have explained abbreviations when needed.

Line 81:”AU” replaced by “Aarhus University”

Milliliter is written as ml (lines 90 and 91) or mL (line 104). Please standardize the use of abbreviations.

It has been changed accordingly and we thank the reviewer for noticing the discrepancy.

Line 117: “mL” replaced by “ml”

Line 113-114: Do you mean the LOQ for both matrices (not for both methods)

We thank the reviewer for mentioning this error. We have changed this in the manuscript.

Line 126: “methods” replaced by “matrices”

Even though the LC-method for cadaverine is described elsewhere by the authors, please indicate here the retention time.

We agree that this would be optimal. However, due to serious illness in the family, the person responsible for this analysis is out of office with no access to data at the moment. We hope that this is not as critical an information to obtain at this moment and that the reviewer will accept to proceed without it. We chose the LC-based MS method due to its high selectivity and sensitivity, and consequently only very small differences would have been observed in retention time.

Quote (Plenis et al. 2019): “The literature data published since 2010 have clearly indicated that LC-based MS techniques have a predominant position in BA investigations due to high selectivity and sensitivity.”

Plenis, A., OlÄ™dzka, I., Kowalski, P., MiÄ™kus, N., & BÄ…czek, T. (2019). Recent Trends in the Quantification of Biogenic Amines in Biofluids as Biomarkers of Various Disorders: A Review. Journal of Clinical Medicine, 8(5). https://doi.org/10.3390/jcm8050640

Results and discussion

Line 171: The LOQ for Rapid’entero and coliform has not been previously defined. The terms LOQ and LOD are not widely used in microbiology to describe the minimum number of microbes that can be detected in a given analytical approach.

We would like to thank the reviewer correcting us on this. We have changed the sentence for better understanding.

Lines 202-203: “were around LOQ” replaced by “were estimates due to low numbers of colonies using the lowest dilution”

Author corrections

Line 59: “portions” added after “The MB”

Line 66: “date” replaced with “day”

Line 203-205: “Brochothrix thermosphacta counts had a 2-log increase for both MB products” replaced by “Counts of Brochothrix thermosphacta in both MB products showed an increase of two logarithmic units during storage”

Line 262: “at D8” added after "decrease”

Line 263: “fall on D8” replaced by “decrease”

Line 277: “for a product at the end of shelf life” added after “standards”

Line 278: “evaluation” replaced by “rejection at D11”

Line 292: “otherwise established through” replaced by "opposed to”
